# Jakiro: Boosting Speculative Decoding with Decoupled Multi-Head via MoE

## Abstract

Speculative decoding (SD) accelerates large language model inference by using a smaller draft model to predict multiple tokens, which are then verified in parallel by the larger target model. However, the limited capacity of the draft model often necessitates tree-based sampling to improve prediction accuracy, where multiple candidates are generated at each step. We identify a key limitation in this approach: the candidates at the same step are derived from the same representation, limiting diversity and reducing overall effectiveness. To address this, we propose Jakiro, leveraging Mixture of Experts (MoE), where independent experts generate diverse predictions, effectively decoupling correlations among candidates. Furthermore, we introduce a hybrid inference strategy, combining autoregressive decoding for initial tokens with parallel decoding for subsequent stages, and enhance the latter with contrastive mechanism in features to improve accuracy. Our method significantly boosts prediction accuracy and achieves higher inference speedups. Extensive experiments across diverse models validate the effectiveness and robustness of our approach, establishing a new SOTA in speculative decoding.

## 1. Introduction

Large language models (LLMs), such as GPT-4o (Jaech et al., 2024) and LLaMA3 (Dubey et al., 2024), have demonstrated remarkable capabilities across a wide range of applications, including question-answering, code synthesis, and machine translation. However, their token-by-token decoding process, combined with the growing size of the models, leads to significant inference latency, posing challenges for real-world deployment. Recently, speculative decoding (Leviathan et al., 2023; Chen et al., 2023) has

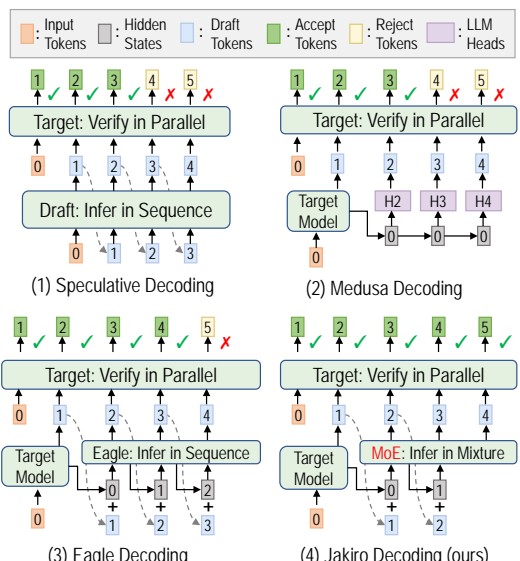

*Figure 1.* Comparison of different speculative decoding methods.

emerged as an effective technique to accelerate LLM inference. This approach utilizes an efficient but weak draft model to predict multiple tokens in sequence, which are then verified in parallel by a more powerful but expensive target model. Since LLMs inference is often memory access bound (Shazeer, 2019), the verification stage can efficiently leverage hardware parallelism, achieving substantial and lossless speedups.

Recent works, such as Medusa (Cai et al., 2024) and Hydra (Ankner et al., 2024), leverage multiple independent heads to predict the next N tokens, with all heads relying on the same final-layer features of the target model. This shared dependency limits the decoupling of predictions for different tokens, thereby constraining the overall prediction accuracy. In contrast, Eagle (Li et al., 2024a) and Eagle2 (Li et al., 2024b) adopts an autoregressive approach, where predictions at different time steps are based on the features from the previous step. Although the Eagle-style approaches successfully decouples draft tokens across different time steps, the top-$k$ tokens within the same layer of the draft tree remain coupled, limiting the diversity of predicted tokens and constraining their immense potential.

To enhance prediction accuracy, both Medusa-style and Eagle-style utilize a tree-based attention mechanism (Miao

---

[1]Anonymous Institution, Anonymous City, Anonymous Region, Anonymous Country. Correspondence to: Anonymous Author <anon.email@domain.com>.

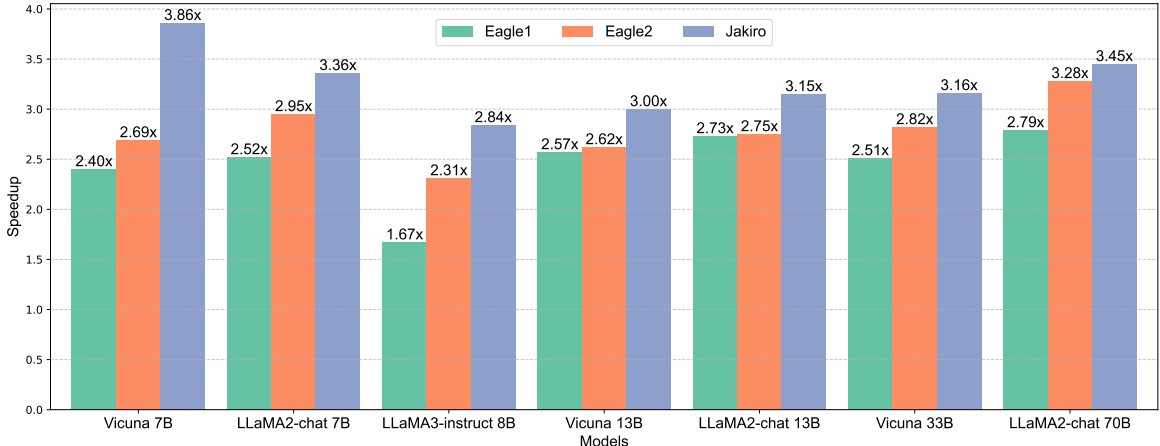

*Figure 2.* Speedup ratio of Vicuna, LLaMA2-chat, and LLaMA3-instruct models inference latency on the MT-bench for non-greedy (Temperature=1) settings. The above reproduction results are based on the open-source code from the original paper and are averaged over four inference runs on an A100-40G GPU. In this paper, we only compare with speculative sampling based methods that do not need to finetune the backbone models, ensuring the output text distribution remains constant.

et al., 2023) to generate multiple candidate tokens simultaneously. However, we observe that these candidates are derived from the same feature representation, resulting in inherent correlations that limit the predictive accuracy of the draft model. To address this, we propose Jakiro with a dynamic decoupling mechanism that leverages an MoE mechanism (Jiang et al., 2024) to construct the attention tree, which retains the independence between layers of the traditional draft tree while introducing decoupling within the layer. By assigning different experts to generate candidate tokens, our method effectively decouples the correlations among predictions. This approach increases the draft model's representation diversity while introducing minimal computational cost. As a result, our method significantly improves the accuracy of the draft model's prediction, leading to a superior overall speedup.

Moreover, due to the inherent exposure bias (Arora et al., 2022) in the autoregressive inference of LLMs, the prediction error increases as the sequence length grows. Considering the inference overhead of the draft model itself, we observe that it is not wise to adopt an autoregressive approach to construct the draft sequence in every step. Instead, we adopt a parallel decoding strategy (Xia et al., 2024) for token generation in the last two steps, further improving the overall inference efficiency of LLMs. To further enhance performance during the parallel decoding phase, we introduce a contrastive mechanism (Li et al., 2022) into MoE's dual-branch heads to improve the helpfulness of predictions, further boosting SD performance in the last two steps. Unlike SCMoE (Shi et al., 2024), which utilizes unselected experts in a self-contrastive manner during inference, we only perform contrastive operations between the two activated expert heads, introducing almost no additional latency. An overview comparison of different speculative decoding

methods is shown in Figure 1.

Compared to existing methods, extensive experiments conducted on common models and benchmarks have validated the effectiveness and robustness of our method. For example, we achieve notable advancements over previous state-of-the-art (SOTA) method on the MT-bench for non-greedy mode, as shown in Figure 2. Our main contributions can be described as:

- **Dynamic Decoupling with MoE in Draft Attention Tree Construction**: This paper is the first to propose leveraging Mixture of Experts (MoE) to achieve dynamic decoupling during the construction of the draft attention tree. This approach enhances the diversity of draft tokens and improves the efficiency of speculative decoding, regardless of whether a static or dynamic tree structure is used.

- **Hybrid Inference Strategy**: We introduce a hybrid inference strategy that combines autoregressive token-by-token prediction for initial tokens with parallel decoding for later stages. During the parallel decoding phase, we innovatively incorporate a contrastive mechanism in the output features to further enhance performance.

- **Comprehensive Benchmark Evaluation**: We conduct comprehensive experiments across various benchmarks and target models to demonstrate the effectiveness and robustness of our approach. Our approach achieves SOTA performance in speculative decoding.

## 2. Preliminaries

### 2.1. Speculative Decoding

Speculative decoding (Leviathan et al., 2023) is designed to accelerate inference from autoregressive LLMs without

altering the output distribution. A smaller, computationally efficient draft model is employed to generate candidate tokens in the drafting phase, while a larger, high-quality target model verifies and refines these candidates. The key idea behind speculative decoding is to parallelize the token generation process, allowing multiple tokens to be proposed at once. This is achieved by observing that complex language-modeling tasks often contain subtasks that can be approximated by smaller models with adequate accuracy. By using these approximations, speculative decoding achieves significant speedup while maintaining the statistical properties of the target model's output.

Suppose $t_i$ denotes the $i$-th token, and let $T_{x:y}$ represent the token sequence $t_x, t_{x+1}, \cdots, t_y$. The speculative decoding process involves three steps:

1. **Drafting:** Given a prefix $T_{1:j}$, the draft model $q(.|.)$ autoregressively generates a sequence of candidate tokens $T_{j+1:j+\gamma}$, each accompanied by a probability distribution $q_{j+1:j+\gamma}$, where $\gamma$ is the number of inference steps per round for the draft model.

2. **Verification:** The target model $p(.|.)$ computes the conditional probabilities $p_{j+1:j+\gamma}$ for the same sequence in a single forward pass. Each candidate token $t_{j+i}$ is sequentially evaluated based on an acceptance criterion, such as $\min(1, p_{j+i}(t_{j+i})/q_{j+i}(t_{j+i}))$.

3. **Refinement:** Accepted tokens are appended to the output sequence, while rejected tokens are resampled using a normalized probability distribution $\text{norm}(\max(0, p_{j+i} - q_{j+i}))$ to replace $t_{j+i}$ and discards the remaining tokens in the drafting.

The method ensures that the output distribution remains consistent with vanilla autoregressive decoding for both greedy and non-greedy sampling strategies, as proven in prior works (Leviathan et al., 2023; Chen et al., 2023).

### 2.2. Medusa-style Decoding

Medusa-style (Cai et al., 2024; Ankner et al., 2024) decoding extends speculative decoding by employing multiple lightweight draft heads, typically implemented as small multi-layer perceptrons (MLPs), atop the last hidden state of the target LLM. These heads independently predict tokens based on the last hidden state of the target LLM, and their predictions are combined into candidate continuations. However, the shared use of the last hidden state across all heads results in incomplete decoupling, limiting the ability of each head to make diverse and independent predictions.

### 2.3. Eagle-style Decoding

In Eagle-style (Li et al., 2024a;b) decoding, the drafting stage decouples draft tokens at different time steps by performing autoregression at the feature level (before the LM

head). The LM head of the original LLM is then used to convert features into draft tokens. This approach reduces uncertainty in the feature sequence, resulting in improved draft predictions. However, the top-$k$ tokens within the same layer of the draft tree remain coupled, which limits the diversity and potential of the predictions.

Eagle1 adopts a static tree-structured draft (Miao et al., 2023) in the verification stage, allowing branching paths to explore alternative continuations when draft tokens are rejected. This structure increases robustness by avoiding the full discard of draft sequences when individual tokens fail to meet the target model's criteria. Eagle2 improves upon Eagle1 by introducing dynamically adjustable draft trees. This allows the tree to adapt based on runtime conditions, optimizing token prediction and verification. Despite this enhancement, the lack of intra-layer decoupling in draft trees still limits the diversity of generated tokens, particularly in non-greedy sampling modes.

## 3. Jakiro

### 3.1. Dynamic Decoupling and MoE Tree Construction

#### 3.1.1. DYNAMIC DECOUPLING WITH MOE HEADS

Unlike Medusa, which relies on the last hidden state for all draft heads, Jakiro employs multiple MoE heads that dynamically allocate expert modules to predict tokens. This mechanism accounts for inherent differences between tokens and ensures the decoupling of token predictions across heads. As a result, prediction confidence is improved while maintaining computational efficiency.

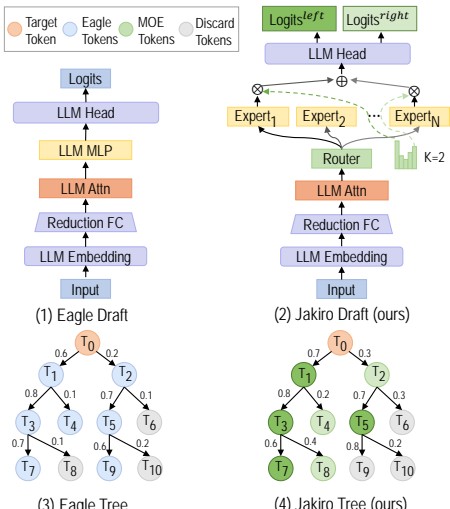

Figure 3. Comparison of different building methods of draft tree.

The structure of our draft model follows the design principles of the Eagle-style, utilizing a single decoder layer that includes a reduction dimension layer, an LLM attention

layer, and parallel expert heads (lightweight MLP layers). And, the embedding layer and head layer remain consistent with the target model without introducing additional parameters. This ensures that inference efficiency is not compromised by the increase in the number of parameters. The detailed structure is shown in Figure 3 (2).

### 3.1.2. MoE Tree Construction

Jakiro introduces a novel tree construction method that decouples the intra-layer dependencies of traditional draft trees while retaining the independence between inter-layers. This design ensures that draft tokens across different layers are generated independently, preserving the consistency of the auto-regressive decoding draft model with the target model and significantly enhancing the accuracy and factuality of predictions. At the same time, the decoupling of intra-layer tokens enables Jakiro to explore a broader range of candidate continuations or diversity, especially in non-greedy modes, without compromising the integrity of the output distribution.

Specifically, the original Eagle inference process can be summarized as follows: when a new token $t_i$ is passed through the embedding layer, a token embedding is obtained. This embedding is then concatenated with the hidden states (denoted as feature $\mathbf{f}_{i-1}$) from the previous step. The concatenated result is processed through the Reduction layer for dimensionality reduction, and the output hidden state $\mathbf{h}_i$ subsequently through the LLM decoder layer (including the attention layer $\mathrm{Attn}$ and MLP layer) to produce the hidden states $\mathbf{f}_i$ for the current step. The next token $t_{i+1}$ is sampled via the Head layer and used for the next step of inference. This process iterates until an end-of-sequence token is encountered or the maximum token length is reached.

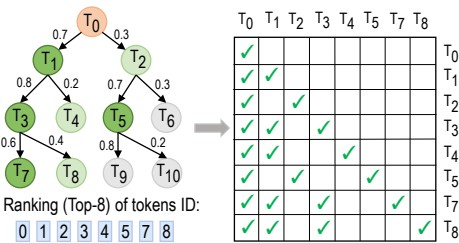

Ranking (Top-8) of tokens ID:
0 1 2 3 4 5 7 8

*Figure 4.* The building process of tree attention mask mechanism.

In contrast, our Jakiro inference process replaces the MLP layer in the LLM with an MoE layer $\mathrm{Expert}_j$ and modifies the draft tree construction. Assuming the number of candidate experts is $N$ and the final number of activated experts is $K$ (default value is 2), the MoE router and the LLM head layer are shared across different expert layers. At each step, two candidate logits distributions are outputed, which are then processed through the MoE Tree decoding.

The logits are sorted based on the scores $s_{j,i}$ corresponding to the selected experts: expert with larger score is placed on the left side of the current layer, while those with smaller score is placed on the right. The formulation is as follows:

$$\mathbf{u}_i = \mathrm{Attn}\left(\mathbf{h}_i\right) + \mathbf{h}_i, \tag{1}$$

$$s_{j,i} = \mathrm{Softmax}_j\left(\mathbf{u}_i^T \mathbf{e}_j\right), \tag{2}$$

$$\mathbf{f}_i = \sum_{j=1}^{N}\left(g_{j,i}\,\mathrm{Expert}_j\left(\mathbf{u}_i\right)\right) + \mathbf{u}_i, \tag{3}$$

$$\mathbf{logits}_i^{\mathrm{left}}, \mathbf{logits}_i^{\mathrm{right}} = \mathrm{Head}\left(s_i^{\mathrm{top1}}\mathbf{f}_i, s_i^{\mathrm{top2}}\mathbf{f}_i\right), \tag{4}$$

$$g_{j,i} = \begin{cases} s_{j,i}, s_{j,i} \in \mathrm{Topk}(\{s_{j,i}|1 \leq j \leq N\}, K), \\ 0, \mathrm{otherwise}, \end{cases} \tag{5}$$

where $\mathbf{e}_j$ is the centroid of the $j$-th expert, the $g_{j,i}$ is sparse, indicating that only $K$ out of $N$ gate values are nonzero. This sparsity property ensures computational efficiency in MoE layer, i.e., each token will be assigned to and computed in only $K$ experts. Also, in the above formulations, we omit the layer normalization operation for brevity. This entire process adheres to the top-$k$ construction principle of the Eagle tree, and the corresponding tree attention mask mechanism is illustrated in Figure 4.

The specific construction process of MoE tree is illustrated in Figure 3 (4). Its implementation mechanism can be combined with the predefined static tree of Eagle1 or the dynamically constructed tree of Eagle2. Experimental results demonstrate that both approaches lead to significant improvements in inference efficiency.

**Impact on Non-Greedy Modes** While traditional speculative decoding methods struggle with diversity in non-greedy sampling, Jakiro excels by introducing mechanisms that maximize both diversity and prediction confidence. This makes it particularly effective in scenarios where alternative continuations need to be explored comprehensively.

### 3.2. Efficient Integration of Contrastive Mechanism

Although Jakiro significantly enhances performance in non-greedy modes through its decoupled tree structure, it does not show a substantial advantage in strictly top-1 accuracy tasks, such as HumanEval and GSM8K, which are optimized for greedy decoding. However, by incorporating the Medusa parallel decoding mechanism, we reduce the number of forward passes during inference without compromising the final speedup, enabling Jakiro to achieve performance on par with Eagle2.

Moreover, LLMs are prone to hallucinations (Tonmoy et al., 2024; Liu et al., 2024), where they generate content that does not align with the facts seen during pretraining. Due to the limited capabilities of the draft model itself, this issue is further exacerbated, making it difficult to further optimize

under greedy decoding (i.e., strict top-1 verification). Although Eagle-style has improved the performance of the draft model by introducing feature distillation, due to the limitations on model parameters, it is hard to gain further improvements through additional training techniques.

However, we explore that the currently popular contrastive decoding technique achieves significant out-of-the-box improvements compared to greedy decoding, without requiring additional training. Previous works (Chuang et al., 2023; O'Brien & Lewis, 2023) have shown that contrastive decoding improves generation quality by searching for strings that maximize the weighted likelihood difference between the strong and weak models. Unlike their use of two highly divergent models in capability, we only apply the contrastive mechanism to the output hidden states, **but not to the logits**, of the two highest-scoring experts in the MoE layer. This approach further improves the generation quality of draft tokens in greedy mode, bringing them closer to the top-1 token of the target model and improving inference speed. The schematic diagram of the mechanism is shown in Figure 5. Below is the detailed derivation of its formulas:

$$\mathbf{f}_i^{\text{moe}} = s_i^{\text{top1}} \mathbf{f}_i^{\text{top1}} + s_i^{\text{top2}} \mathbf{f}_i^{\text{top2}}, \quad (6)$$

$$\mathbf{f}_i^{\text{const}} = \beta \mathbf{f}_i^{\text{top1}} - \alpha \mathbf{f}_i^{\text{top2}}, \quad (7)$$

$$\mathbf{logits}_i^{\text{moe}}, \mathbf{logits}_i^{\text{const}} = \text{Head}\left(\mathbf{f}_i^{\text{moe}}, \mathbf{f}_i^{\text{const}}\right), \quad (8)$$

where the $\mathbf{logits}_i^{\text{moe}}$ indicates that the inference is based on the logits output from the previous $[1 : \gamma - 1]$ steps for autoregression decoding, while $\mathbf{logits}_i^{\text{const}}$ is used for parallel decoding at step $\gamma - 1$, resulting in one fewer draft process compared to the Eagle-style method. Additionally, $\beta$ and $\alpha$ are learnable parameters corresponding to the scores of the candidate experts being top1 and top2, respectively, and are used to adaptively adjust the difference between them for better contrastive learning. Compared to the non-greedy mode of Jakiro, no additional parameters are introduced, meaning **the same weights** are used.

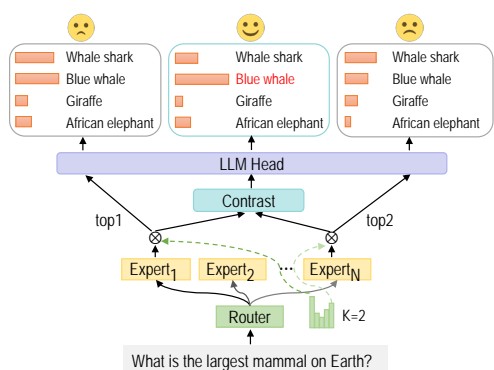

*Figure 5.* Efficient integration of contrastive mechanism.

Additionally, it should be noted that our Jakiro is not limited to parallel decoding for the last two steps. When $K > 2$, it

theoretically supports parallel decoding for the last $K$ steps. However, this is not the focus of the current study and will be explored in future work.

### 3.3. Training of the Draft Models

Similar to Eagle-style, in order to reduce training costs, we use a preprocessed fixed dataset for training the draft models. Other data augmentation hyperparameters are kept consistent with Eagle. Additionally, we adopt the Smooth L1 loss for predicting the next feature as a regression task and use cross-entropy to ensure the accuracy of tokens sequence. However, considering that the entire process of the draft model is autoregressive and computationally expensive, we use a parallel decoding mechanism during the penultimate step. Thus, the prediction of the next token is achieved through contrast between the two expert heads of the MoE, introducing an additional loss (similar to the Medusa implementation). The optimize objective is as follows:

$$L_{reg}^{\text{moe}} = \text{Smooth L1}(\mathbf{f}_{i+1}^p, \mathbf{f}_i^{\text{moe}}), \quad (9)$$

$$L_{reg}^{\text{const}} = \text{Smooth L1}(\mathbf{f}_{i+2}^p, \mathbf{f}_i^{\text{const}}), \quad (10)$$

$$q_{i+2}, q_{i+3} = \text{Softmax}(\mathbf{logits}_i^{\text{moe}}, \mathbf{logits}_i^{\text{const}}), \quad (11)$$

$$L_{cls}^{\text{moe}} = \text{Cross\_Entropy}(p_{i+2}, q_{i+2}), \quad (12)$$

$$L_{cls}^{\text{const}} = \text{Cross\_Entropy}(p_{i+3}, q_{i+3}), \quad (13)$$

$$L = L_{reg}^{\text{moe}} + w_{cls}^{\text{moe}} L_{cls}^{\text{moe}} + L_{reg}^{\text{const}} + w_{cls}^{\text{const}} L_{cls}^{\text{const}} \quad (14)$$

We use a combined loss function $L$ to train the autoregressive draft model. Given that the classification loss is an order of magnitude larger than the regression loss numerically, and the importance of const is lower compared to moe, we set $w_{cls}^{\text{moe}}$ and $w_{cls}^{\text{const}}$ to 0.1 and 0.05, respectively.

## 4. Experiments

**Models and tasks.** Following the current mainstream works, we conduct experiments on Vicuna models (7B, 13B, 33B) (Chiang et al., 2023), LLaMA2-chat models (7B, 13B, 70B) (Touvron et al., 2023), and LLaMA3-Instruct (8B, 70B) (Meta, 2024), encompassing the common sizes of current mainstream LLMs. To evaluate the generality and robustness of our method, our Jakiro is evaluated across multiple tasks including multi-turn dialogue, code generation, mathematical reasoning, and instruction following, employing the MT-bench (Zheng et al., 2023), HumanEval (Chen et al., 2021), GSM8K (Cobbe et al., 2021), Alpaca (Taori et al., 2023), CNN/Daily Mail (Nallapati et al., 2016), and Natural Questions (Kwiatkowski et al., 2019) datasets, respectively. Because Speculative decoding (Leviathan et al., 2023) conduct experiments with a batch size of 1, a setting subsequently adopted by other works such as Medusa-style (Cai et al., 2024; Ankner et al., 2024) and Eagle-style (Li et al., 2024a;b). Similarly, the majority of our experiments also adopt this setting.

**Metrics.** Similar to prior speculative sampling approaches, our Jakiro method primarily focuses on reducing latency rather than optimizing throughput. To measure its acceleration effects, we utilize the following metrics:

- **Walltime speedup ratio:** Compared to traditional autoregressive decoding, the ratio of speedup achieved in actual tests of end-to-end.

- **Average acceptance length $\tau$:** The average number of tokens accepted from speculative decoding per forward pass of the target LLM.

Similar to methods that use a strict speculative sampling mechanism, the acceleration achieved by Jakiro ensures that the output distribution of the target LLMs is maintained. Therefore, evaluating the quality of the generated results is both unnecessary and irrelevant, as the focus is on efficiency rather than output fidelity.

**Training.** We keep the target LLMs fixed throughout the training process. Our proposed Jakiro model is trained on the ShareGPT[1] dataset using 68,000 dialogue iterations without needing extensive retraining or additional data beyond the pre-trained models, and with a learning rate set at 9e-5. The AdamW optimizer is employed with beta values $(\beta_1, \beta_2) = (0.9, 0.95)$, and gradient clipping is applied with a threshold of 0.5. The number of trainable parameters for Jakiro varies across model sizes: 0.35B for the 7B model, 0.56B for the 8B model, 0.88B for the 13B model, 1.42B for the 33B model, and 1.87B for the 70B model. For example, training the Jakiro draft model for the 70B model takes approximately 2-3 days on 4×A100 40GB GPU.

**Testing.** Our experiments are conducted on different devices (AMD Instinct$^{\text{TM}}$ MI250-64G, NVIDIA A40-45G, and NVIDIA A100-40G) and the limitation of hardware GPU memory. For models of size (7B, 8B, 13B), (33B), and (77B), we perform single-GPU, two GPUs, and four GPUs, respectively.

### 4.1. Effectiveness

As shown in Table 1 and Table 2, our proposed Jakiro consistently outperforms existing methods in terms of speedup ratios across various datasets and models on MI250. The speedup ratios achieved by Jakiro, such as 2.99x on the MT-bench and 3.43x on the HumanEval task for the Vicuna 7B model under greedy mode, significantly surpass those of other methods, including Eagle2, which achieved a maximum speedup of 2.88x on average. The results across different tasks further highlight the effectiveness of Jakiro. For example, on the GSM8K task, Jakiro achieves a 3.11x speedup, maintaining a high average acceptance length of 4.95 tokens. This demonstrates that Jakiro not only acceler-

---
[1]We can download ShareGPT from Huggingface at link.

ates inference but also retains high-quality output, maintaining the target LLM's performance without compromising accuracy.

Since our Jakiro involves one less drafting step than Eagle2, there is no significant improvement in the average acceptance length of tokens. However, considering that the ultimate goal of speculative decoding is to improve speedup, our method is still effective. For instance, in the greedy mode for LLaMA2-Chat 7B, although the average acceptance length of our method is slightly lower than that of Eagle2 (4.51 vs. 4.63), the final speedup of our method is notably improved (2.83x vs. 2.66x). Furthermore, we observe that Jakiro excels across both code generation and natural language tasks. This is particularly notable because code generation tasks often benefit from greedy sampling, where the inherent structure of code makes it easier to predict subsequent tokens efficiently. Similarly, on Natural Questions and summarization tasks, Jakiro maintains superior performance.

Additionally, in non-greedy mode, the comparison methods with speculative sampling also reveal that Jakiro achieves significantly higher speedup ratios while also maintaining a higher average acceptance length. Such as Vicuna 7B, compared to Eagle2, our method achieves a 15.4% improvement in the final average speedup across all benchmark test sets (2.84 vs. 2.46). This suggests that Jakiro benefits from a more efficient drafting process that allows for longer and more stable sequences of tokens to be accepted, reducing the need for frequent re-sampling and minimizing the risk of errors during the inference process.

### 4.2. Ablation Study

**N-k Setting of MoE.** The number of candidate experts $N$ and activated experts $K$ involved in the computation for each token in a typical MoE architecture is 8 and 2, respectively. If speculative decoding uses the original chain-based construction of draft tokens, the number of experts activated each time is fixed and equal to 2. However, current mainstream speculative decoding methods use a tree structure to construct draft tokens. Although the number of candidate experts activated for each token is fixed, the process of constructing the tree results in the top-k candidate tokens being selected at each layer, which leads to a number of experts activated K$\geq$ 2 during each inference. Additionally, the more candidate experts there are, the better the model's predictive ability, but with an increased computational cost. Therefore, it is necessary to choose an appropriate N-K setting to achieve optimal inference speed. The results under different N-K settings are shown in Table 3.

From the above results, it can be seen that as the number of candidate experts increases, both the average acceptance length and the computational load increase accordingly.

*Table 1.* Speedup ratios and average acceptance lengths $\tau$ of different methods on MI250. V represents Vicuna, L2 represents LLaMA2-Chat, and L3 represents LLaMA3-Instruct. SpS denotes standard speculative sampling, with its draft model being Vicuna-68M. Methods like Medusa relax acceptance conditions under non-greedy settings, which do not guarantee lossless acceleration. Therefore, we do not compare Jakiro with these methods. Where the symbol * indicates that our method uses MoE-based weighted decoupling to construct the draft tree, as shown in Figure 3(2). Without the symbol *, it refers to the construction of the draft tree using weighted non-decoupling combined with the contrastive mechanism, as shown in Figure 5. The results on other devices can be found in the appendix.

| Model | Method | MT-bench | | HumanEval | | GSM8K | | Alpaca | | CNN/DM | | Natural Ques. | | Mean | |
|---|---|---|---|---|---|---|---|---|---|---|---|---|---|---|---|
| | | Speedup | $\tau$ | Speedup | $\tau$ | Speedup | $\tau$ | Speedup | $\tau$ | Speedup | $\tau$ | Speedup | $\tau$ | Speedup | $\tau$ |
| | | Temperature=0 | | | | | | | | | | | | | |
| V 7B | SpS | 1.82x | 2.36 | 1.99x | 2.61 | 1.71x | 2.26 | 1.65x | 2.21 | 1.81x | 2.44 | 1.60x | 2.16 | 1.76x | 2.34 |
| | Medusa | 1.91x | 2.52 | 2.02x | 2.67 | 1.89x | 2.59 | 1.79x | 2.48 | 1.42x | 2.02 | 1.51x | 2.09 | 1.76x | 2.40 |
| | Hydra | 2.69x | 3.60 | 2.98x | 3.79 | 2.73x | 3.66 | 2.66x | 3.58 | 2.01x | 2.70 | 2.25x | 2.86 | 2.55x | 3.37 |
| | Eagle1 | 2.52x | 3.97 | 2.82x | 4.30 | 2.58x | 4.01 | 2.37x | 3.87 | 2.15x | 3.43 | 2.05x | 3.22 | 2.42x | 3.80 |
| | Eagle2 | 2.75x | 4.94 | 3.12x | 5.35 | 2.81x | 4.94 | 2.63x | **4.85** | 2.25x | 4.11 | 2.17x | 3.84 | 2.62x | 4.67 |
| | Jakiro | **2.99x** | **4.96** | **3.43x** | **5.36** | **3.11x** | **4.95** | **2.87x** | 4.82 | **2.50x** | **4.20** | **2.38x** | 3.84 | **2.88x** | **4.69** |
| L2 7B | Eagle2 | 2.70x | **4.70** | 3.12x | **5.38** | 2.78x | **4.76** | 2.67x | **4.65** | 2.27x | **4.09** | 2.41x | **4.16** | 2.66x | **4.63** |
| | Jakiro | **2.89x** | 4.61 | **3.32x** | 5.26 | **2.97x** | 4.66 | **2.81x** | 4.47 | **2.42x** | 4.01 | **2.56x** | 4.07 | **2.83x** | 4.51 |
| V 13B | Eagle2 | 3.02x | **4.84** | 3.51x | **5.42** | 3.11x | **4.82** | 2.95x | **4.89** | 2.60x | **4.28** | 2.37x | 3.69 | 2.93x | **4.66** |
| | Jakiro | **3.18x** | 4.74 | **3.70x** | 5.36 | **3.30x** | 4.81 | **3.03x** | 4.68 | **2.69x** | 4.20 | **2.52x** | 3.70 | **3.07x** | 4.58 |
| L2 13B | Eagle2 | 3.06x | **4.74** | 3.62x | **5.53** | 3.19x | 4.88 | 2.96x | 4.62 | 2.66x | 4.24 | 2.68x | 4.12 | 3.03x | 4.69 |
| | Jakiro | **3.22x** | 4.72 | **3.76x** | 5.41 | **3.41x** | **4.96** | **3.18x** | **4.67** | **2.83x** | **4.32** | **2.87x** | **4.17** | **3.21x** | **4.71** |
| | | Temperature=1 | | | | | | | | | | | | | |
| V 7B | SpS | 1.50x | 1.87 | 1.55x | 1.95 | 1.53x | 1.82 | 1.56x | 1.85 | 1.63x | 1.91 | 1.33x | 1.72 | 1.52x | 1.85 |
| | Eagle1 | 2.18x | 3.59 | 2.37x | 3.82 | 2.16x | 3.56 | 2.12x | 3.72 | 1.88x | 3.10 | 1.78x | 2.91 | 2.08x | 3.45 |
| | Eagle2 | 2.39x | 4.26 | 2.64x | 4.67 | 2.37x | 4.49 | 2.28x | 4.28 | 2.01x | 3.77 | 1.95x | 3.54 | 2.27x | 4.17 |
| | Jakiro | 2.50x | 4.22 | **2.92x** | **4.70** | 2.60x | 4.44 | 2.48x | 4.28 | 2.16x | 3.77 | 2.08x | 3.49 | 2.46x | 4.15 |
| | Jakiro* | **3.18x** | **5.65** | 2.48x | 4.52 | **2.91x** | **5.50** | **2.99x** | **5.67** | **2.70x** | **5.33** | **2.80x** | **5.05** | **2.84x** | **5.32** |
| L2 7B | Eagle2 | 2.61x | 4.48 | 2.86x | 5.04 | 2.67x | **4.72** | 2.48x | 4.37 | 2.14x | 3.92 | 2.30x | 4.06 | 2.51x | 4.43 |
| | Jakiro | 2.68x | 4.45 | **3.04x** | **4.94** | **2.82x** | 4.61 | 2.64x | 4.32 | 2.27x | 3.87 | 2.44x | 3.98 | 2.65x | 4.36 |
| | Jakiro* | **2.93x** | **5.16** | 2.62x | 4.73 | 2.70x | 4.59 | **3.04x** | **5.22** | **2.99x** | **5.37** | **2.73x** | **4.81** | **2.84x** | **4.98** |
| V 13B | Eagle2 | 2.72x | 4.30 | 3.09x | 4.78 | 2.85x | 4.52 | 2.63x | 4.33 | 2.38x | 3.94 | 2.22x | 3.57 | 2.65x | 4.24 |
| | Jakiro | **2.84x** | **4.27** | **3.31x** | **4.95** | **3.00x** | **4.50** | **2.73x** | **4.31** | **2.45x** | **3.88** | **2.36x** | **3.54** | **2.78x** | **4.24** |
| L2 13B | Eagle2 | 2.90x | 4.58 | 3.45x | 5.37 | 3.04x | 4.73 | 2.86x | 4.44 | 2.56x | 4.21 | 2.62x | 4.04 | 2.90x | **4.56** |
| | Jakiro | **3.06x** | 4.49 | **3.60x** | 5.22 | **3.26x** | **4.76** | **3.01x** | 4.43 | **2.71x** | 4.15 | **2.83x** | 4.13 | **3.08x** | 4.53 |
| V 33B | Eagle2 | 3.04x | 4.22 | 3.52x | 4.78 | 3.41x | 4.55 | 2.87x | 3.99 | 2.56x | 3.71 | 2.40x | 3.27 | 2.97x | 4.09 |
| | Jakiro | **3.16x** | **4.31** | **3.64x** | **4.84** | **3.52x** | **4.63** | **2.98x** | **4.02** | **2.68x** | **3.85** | **2.52x** | **3.38** | **3.08x** | **4.17** |
| L2 70B | Eagle2 | 2.96x | 4.48 | 3.49x | 5.18 | 3.09x | 4.59 | 2.97x | 4.49 | 2.34x | 3.64 | 2.54x | 3.84 | 2.90x | 4.37 |
| | Jakiro | 3.08x | 4.56 | **3.62x** | **5.25** | 3.18x | 4.61 | 3.05x | 4.47 | 2.53x | 3.87 | 2.65x | 3.90 | 3.02x | 4.44 |
| | Jakiro* | **3.68x** | **5.16** | 3.18x | 4.37 | **3.43x** | **4.71** | **3.60x** | **4.90** | **3.55x** | 5.18 | **3.56x** | **4.86** | **3.50x** | **4.86** |
| L3 8B | Eagle2 | 2.31x | 3.97 | 2.96x | 4.83 | 2.65x | 4.36 | 2.52x | 4.50 | 1.99x | 3.52 | 2.05x | 3.36 | 2.41x | 4.09 |
| | Jakiro | **2.58x** | **4.66** | **3.02x** | **4.91** | **2.70x** | **4.44** | **2.64x** | **4.53** | **2.06x** | **3.59** | **2.32x** | **3.95** | **2.55x** | **4.35** |
| L3 70B | Eagle2 | 3.11x | 4.98 | 2.83x | 4.35 | 2.73x | 4.67 | 2.10x | 3.64 | 2.34x | 3.52 | 2.30x | 3.42 | 2.57x | 4.10 |
| | Jakiro | **3.26x** | **5.04** | **2.89x** | **4.42** | **2.91x** | **4.77** | **2.46x** | **3.78** | **2.35x** | **3.75** | **2.32x** | **3.56** | **2.70x** | **4.22** |

*Table 2.* Speedup ratios and average acceptance lengths $\tau$ with other models as the original LLMs, with the temperature set to 0.

| Model | Method | MT-bench | | HumanEval | | GSM8K | |
|---|---|---|---|---|---|---|---|
| | | Speedup | $\tau$ | Speedup | $\tau$ | Speedup | $\tau$ |
| LLaMA3-Instruct 8B | Eagle1 | 1.84x | 3.07 | 2.27x | 3.72 | 2.25x | 3.70 |
| | Eagle2 | 2.63x | **4.32** | 3.08x | **5.06** | 2.79x | **4.46** |
| | Jakiro | **2.74x** | 4.23 | **3.25x** | 5.01 | **2.92x** | 4.38 |
| Vicuna 33B | Eagle1 | 2.87x | 3.69 | 3.41x | 4.28 | 3.16x | 3.93 |
| | Eagle2 | 3.21x | **4.43** | 3.89x | **5.20** | 3.52x | 4.66 |
| | Jakiro | **3.34x** | 4.40 | **3.96x** | 5.16 | **3.67x** | **4.67** |
| LLaMA2-Chat 70B | Eagle1 | 2.83x | 3.84 | 3.33x | 4.44 | 2.95x | 3.92 |
| | Eagle2 | 2.98x | **4.51** | 3.54x | **5.25** | 3.10x | **4.63** |
| | Jakiro | **3.01x** | 4.46 | **3.58x** | 5.19 | **3.16x** | 4.58 |
| LLaMA3-Instruct 70B | Eagle1 | 2.43x | 3.33 | 3.01x | 4.15 | 2.53x | 3.87 |
| | Eagle2 | 2.58x | **4.14** | 3.10x | **5.09** | 2.82x | **4.34** |
| | Jakiro | **2.65x** | 4.06 | **3.28x** | 5.03 | **2.97x** | 4.29 |

*Table 3.* Ablation experiment results with temperature set to 0 on Vicuna 7B under the MI250. "N" indicates the candidate experts, and "K" indicates the experts activated each time.

| N | K | MT-bench | | HumanEval | | GSM8K | | Alpaca | |
|---|---|---|---|---|---|---|---|---|---|
| | | Speedup | $\tau$ | Speedup | $\tau$ | Speedup | $\tau$ | Speedup | $\tau$ |
| 5 | 2 | 2.67x | **5.13** | 3.03x | **5.59** | 2.71x | **5.14** | 2.56x | **5.10** |
| 4 | 2 | 2.69x | 5.09 | 3.07x | 5.54 | 2.75x | 5.12 | 2.57x | 5.04 |
| 3 | 2 | 2.75x | 5.09 | 3.13x | 5.54 | 2.79x | 5.04 | 2.61x | 5.04 |
| 2 | 2 | **2.92x** | 4.97 | **3.36x** | 5.45 | **3.01x** | 4.97 | **2.83x** | 4.98 |

However, we believe that combining hardware and algorithmic optimizations can mitigate these costs, further unlocking the potential of MoE, which remains an area for future exploration.

**Combine with Parallel Decoding and Contrastive Mechanism.** The contrastive mechanism (perhaps similar to a form of contrastive learning) and its implementation combined with parallel decoding is discussed in Section 3.2. This approach aims to reduce one draft inference step compared to conventional autoregressive decoding LLMs (e.g.,

Eagle2) without significant loss in prediction performance, thereby achieving optimal speedup. However, the essence of these two mechanisms is different. The contrastive mechanism focuses on better predicting the next token in the sequence, while parallel decoding aims to output multiple future tokens in one step. Therefore, it is necessary to conduct further experiments to evaluate their respective impacts on final performance, with the results shown in Table 4.

As can be seen from the above results, using parallel decoding or the contrastive mechanism alone does lead to some improvement compared to using MoE alone, which demonstrates that the two proposed mechanisms can effectively complement Jakiro. However, combining both mechanisms further unleashes the potential of Jakiro, maintaining a comparable acceptance rate (similar to average accpet length) while further improving its speedup.

*Table 4.* Ablation experiment results with temperature set to 0 on Vicuna 7B under the MI250. "PD" refers to whether the penultimate inference in the draft process uses the parallel decoding mechanism. "Const" indicates whether contrastive mechanism on features is applied in the penultimate inference of the draft process.

| | MT-bench | | HumanEval | | GSM8K | | Alpaca | |
|---|---|---|---|---|---|---|---|---|
| Method | Speedup | $\tau$ | Speedup | $\tau$ | Speedup | $\tau$ | Speedup | $\tau$ |
| w/o both | 2.92x | 4.97 | 3.36x | 5.45 | 3.01x | 4.97 | 2.83x | 4.98 |
| w/o PD | 2.98x | **5.04** | 3.41x | **5.50** | 3.10x | **5.11** | 2.83x | **4.92** |
| w/o Const | 2.98x | 4.86 | 3.42x | 5.33 | 3.05x | 4.86 | 2.86x | 4.82 |
| Jakiro | **2.99x** | 4.96 | **3.43x** | 5.36 | **3.11x** | 4.95 | **2.87x** | 4.82 |

## 5. Related Work

**Speculative decoding:** Speculative decoding (SD)has emerged as a powerful technique to accelerate LLMs inference by reducing memory bandwidth bottlenecks. Early speculative decoding methods, such as those by (Stern et al., 2018) and (Sun et al., 2021), focused on greedy decoding strategies, while (Leviathan et al., 2023) and (Chen et al., 2023) expanded speculative sampling to non-greedy decoding. Recent works in SD have enhanced draft model efficiency, with methods like SpecInfer (Miao et al., 2023) employing tree attention to verify multiple draft tokens in parallel, and Medusa (Cai et al., 2024) using extra MLP heads to generate token drafts. Although these methods have achieved notable acceleration, they face limitations in token diversity and decoupling between draft and target models. Eagle (Li et al., 2024a) introduces a more dynamic approach by decoupling the draft tokens across different time steps. However, Eagle still maintains coupling between the Top-k tokens at the same layer in the draft tree, which restricts the diversity and specialization of tokens. Our approach builds upon these limitations by introducing a dynamic decoupling mechanism with Mixture of Experts (MoE) heads, which enables draft tokens to consider inherent differences between them, leading to more diversity and confident predictions.

**Mixture of Experts:** MoE has been explored extensively

for improving the specialization of models. MoE techniques were originally proposed by (Jacobs et al., 1991; Jordan & Jacobs, 1994) and later adapted to language models, such as in GShard (Lepikhin et al., 2021) and Switch Transformer (Fedus et al., 2021), which scale MoE to large models with top-k routing strategies. More recently, the integration of MoE in Transformer-based architectures has garnered significant interest, with methods like StableMoE (Dai et al., 2022) exploring fixed routing strategies for more stable training. MoE heads have also been used in multi-modal settings (Du et al., 2022; Xue et al., 2023), where they enable specialization across different modalities. Furthermore, in the context of speculative decoding, our method combines MoE's dual-branch heads with contrastive decoding techniques, a strategy inspired by recent works (Shazeer et al., 2017; Dai et al., 2022) to improve the usefulness of draft token predictions, especially in greedy modes. By integrating these strategies, we can achieve more reliable predictions with faster inference, as demonstrated through our experiments.

**Parallel decoding:** Parallel decoding is known for its efficiency in machine translation (Ghazvininejad et al., 2019) and code generation (Gloeckle et al., 2024), has also been integrated into SD frameworks to further enhance efficiency. Although the use of parallel decoding in speculative frameworks has been under-explored, works like (Monea et al., 2023) and (Yi et al., 2024) have pioneered its application. These methods, however, still face challenges in achieving a perfect alignment between draft distributions and target models, which can limit their effectiveness in lossless acceleration. Our approach addresses these challenges by improving the decoupling mechanism within the MoE heads, ensuring better alignment and more diverse token predictions in both greedy and non-greedy modes.

## 6. Conclusion

In summary, while speculative decoding has proven effective for accelerating LLM inference, challenges remain in improving token diversity and maintaining distribution alignment between draft and target models. Our work makes significant strides in addressing these challenges by introducing a dynamic decoupling mechanism with MoE heads, which improves prediction confidence, diversity, and factuality. Additionally, by combining contrastive mechanism with MoE's dual-branch heads, we further enhance the utility of our predictions, demonstrating the potential of this approach in both greedy and non-greedy generation tasks. Our proposed Jakiro demonstrates superior speedup ratios, high acceptance lengths, and overall efficiency across a variety of tasks and model sizes, making it a compelling solution for improving inference speed without sacrificing model output quality.

## Impact Statement

This paper presents work whose goal is to advance the field of Machine Learning. There are many potential societal consequences of our work, none which we feel must be specifically highlighted here.

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

# A. Supplementary experiments

## A.1. Average Acceptance Length.

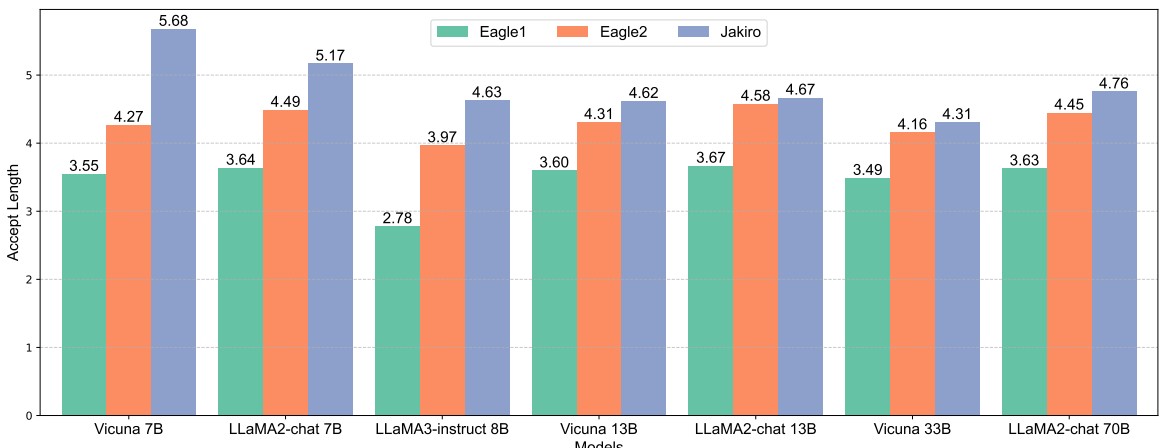

*Figure 6.* Average acceptance length of Vicuna, LLaMA2-chat, and LLaMA3-instruct models inference on the MT-bench for non-greedy (Temperature=1) settings. The above reproduction results are based on the open-source code from the original paper and are averaged over four inference runs on an A100-40G GPU. In this paper, we only compare with speculative sampling based methods that do not need to finetune the backbone models, ensuring the output text distribution remains constant.

From Figure 6, it can be observed that our Jakiro achieves the highest acceptance rate (approximately average acceptance length) across different models, further demonstrating the robustness of our method. Under the advantage of the MoE mechanism, it fully utilizes the diversity prediction of draft tokens in the non-greedy mode, resulting in an increased number of accepted tokens and thus improving the average length accepted by the target model during the validation phase.

## A.2. Results of Other Devices.

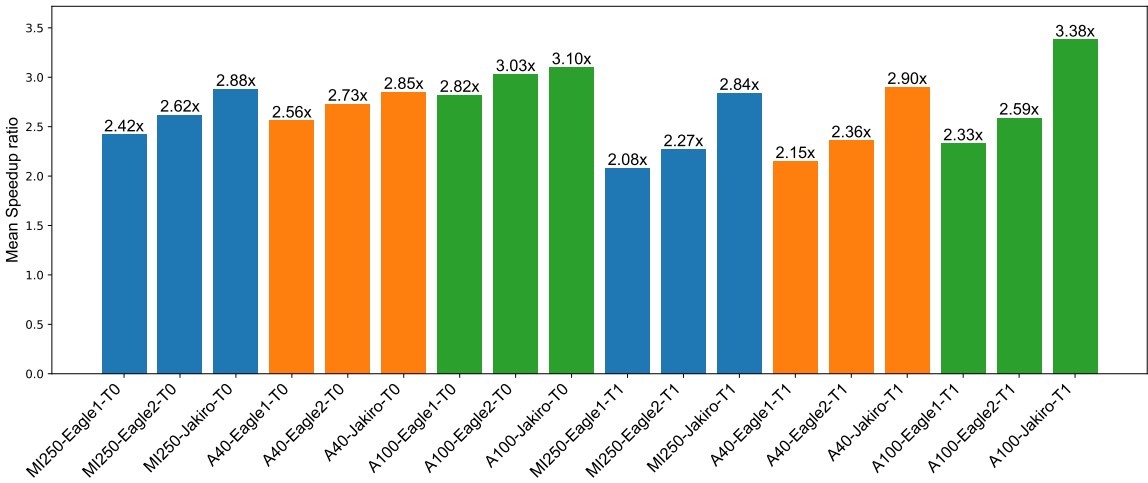

*Figure 7.* Speedup ratios of Vicuna 7B inference on all tasks under different devices. Where the suffix "T0" represents greedy mode sampling, while suffix "T1" represents non-greedy sampling. The above reproduction results are based on the open-source code from the original paper and are averaged over four inference runs. In this paper, we only compare with speculative sampling based methods that do not need to finetune the backbone models, ensuring the output text distribution remains constant.

As shown Figure 7, our proposed Jakiro achieves consistent improvements across different devices, indicating that our method is robust and has the capability of performance transferability across devices.

**Results in A40-45G:**

*Table 5.* Speedup ratios and average acceptance lengths $\tau$ of different methods. V represents Vicuna, L2 represents LLaMA2-Chat. SpS denotes standard speculative sampling, with its draft model being Vicuna-68M. Methods like Medusa relax acceptance conditions under non-greedy settings, which do not guarantee lossless acceleration. Therefore, we do not compare Jakiro with these methods. Where the symbol * indicates that our method uses MoE-based weighted decoupling to construct the draft tree, as shown in Figure 3(2). Without the symbol *, it refers to the construction of the draft tree using weighted non-decoupling combined with the contrastive mechanism, as shown in Figure 5.

| Model | Method | MT-bench Speedup | $\tau$ | HumanEval Speedup | $\tau$ | GSM8K Speedup | $\tau$ | Alpaca Speedup | $\tau$ | CNN/DM Speedup | $\tau$ | Natural Ques. Speedup | $\tau$ | Mean Speedup | $\tau$ |
|---|---|---|---|---|---|---|---|---|---|---|---|---|---|---|---|
| | | | | | | | Temperature=0 | | | | | | | | |
| V 7B | Eagle1 | 2.69x | 3.98 | 2.98x | 4.29 | 2.74x | 4.00 | 2.56x | 3.87 | 2.21x | 3.42 | 2.17x | 3.21 | 2.56x | 3.80 |
| | Eagle2 | 2.92x | 4.98 | 3.24x | 5.34 | 2.94x | 4.95 | 2.76x | **4.87** | 2.28x | 4.12 | 2.26x | 3.81 | 2.73x | 4.68 |
| | Jakiro | **3.02x** | **4.98** | **3.40x** | **5.37** | **3.08x** | **4.97** | **2.86x** | 4.82 | **2.38x** | **4.17** | **2.36x** | **3.83** | **2.85x** | **4.69** |
| L2 7B | Eagle1 | 2.62x | 3.82 | 2.94x | 4.30 | 2.66x | 3.90 | 2.53x | 3.70 | 2.24x | 3.41 | 2.34x | 3.44 | 2.55x | 3.76 |
| | Eagle2 | 2.81x | **4.71** | 3.23x | **5.39** | 2.86x | **4.76** | 2.79x | **4.66** | 2.31x | **4.12** | 2.51x | **4.19** | 2.75x | **4.64** |
| | Jakiro | **2.87x** | 4.61 | **3.29x** | 5.25 | **2.93x** | 4.64 | **2.81x** | 4.47 | **2.32x** | 4.01 | **2.56x** | 4.07 | **2.80x** | 4.51 |
| V 13B | Eagle1 | 2.90x | 4.00 | 3.28x | 4.39 | 2.94x | 3.97 | 2.74x | 3.95 | 2.45x | 3.52 | 2.28x | 3.11 | 2.77x | 3.82 |
| | Eagle2 | 2.97x | **4.83** | 3.45x | **5.41** | 3.03x | 4.79 | 2.89x | **4.89** | 2.46x | **4.22** | 2.33x | **3.74** | 2.86x | **4.65** |
| | Jakiro | **3.11x** | 4.76 | **3.61x** | 5.36 | **3.22x** | **4.81** | **2.96x** | 4.69 | **2.49x** | 4.13 | **2.45x** | 3.67 | **2.97x** | 4.57 |
| L2 13B | Eagle1 | 2.89x | 3.93 | 3.39x | 4.52 | 2.99x | 4.03 | 2.81x | 3.83 | 2.54x | 3.59 | 2.58x | 3.47 | 2.87x | 3.89 |
| | Eagle2 | 2.96x | **4.74** | 3.52x | **5.52** | 3.09x | 4.90 | 2.88x | 4.60 | 2.53x | 4.25 | 2.60x | 4.11 | 2.93x | 4.69 |
| | Jakiro | **3.11x** | 4.73 | **3.62x** | 5.42 | **3.30x** | **4.94** | **3.07x** | **4.65** | **2.62x** | **4.30** | **2.79x** | **4.16** | **3.08x** | **4.70** |
| | | | | | | | Temperature=1 | | | | | | | | |
| V 7B | Eagle1 | 2.23x | 3.54 | 2.41x | 3.82 | 2.23x | 3.66 | 2.23x | 3.63 | 1.91x | 3.13 | 1.86x | 2.95 | 2.15x | 3.45 |
| | Eagle2 | 2.46x | 4.29 | 2.72x | 4.63 | 2.47x | 4.41 | 2.41x | 4.47 | 2.09x | 3.82 | 2.03x | 3.50 | 2.36x | 4.19 |
| | Jakiro | 2.54x | 4.15 | **2.95x** | **4.72** | 2.63x | 4.48 | 2.54x | 4.47 | 2.19x | 3.89 | 2.12x | 3.51 | 2.50x | 4.20 |
| | Jakiro* | **3.24x** | **5.64** | 2.67x | 4.12 | **2.84x** | **5.76** | **2.97x** | **5.67** | **2.84x** | **5.59** | **2.85x** | **5.04** | **2.90x** | **5.47** |
| L2 7B | Eagle1 | 2.24x | 3.59 | 2.55x | 4.08 | 2.39x | 3.80 | 2.21x | 3.51 | 1.96x | 3.18 | 2.04x | 3.24 | 2.23x | 3.56 |
| | Eagle2 | 2.63x | 4.53 | 2.97x | 5.04 | 2.79x | 4.76 | 2.63x | 4.53 | 2.18x | 3.93 | 2.34x | 3.99 | 2.59x | 4.46 |
| | Jakiro | 2.66x | 4.45 | **2.99x** | **4.93** | 2.80x | 4.59 | 2.66x | 4.40 | 2.19x | 3.93 | 2.37x | 3.89 | 2.61x | 4.37 |
| | Jakiro* | **2.97x** | **5.13** | 2.70x | 4.68 | **2.80x** | **4.69** | **3.06x** | **5.19** | **2.97x** | **5.46** | **2.80x** | **4.85** | **2.88x** | **5.00** |
| V 13B | Eagle1 | 2.51x | 3.66 | 2.84x | 4.02 | 2.58x | 3.74 | 2.49x | 3.77 | 2.22x | 3.32 | 2.11x | 3.03 | 2.46x | 3.59 |
| | Eagle2 | 2.68x | 4.38 | 3.03x | 4.86 | 2.69x | 4.46 | 2.53x | 4.48 | 2.24x | 3.91 | 2.15x | 3.53 | 2.55x | 4.27 |
| | Jakiro | 2.78x | 4.23 | 3.28x | 4.89 | 2.90x | 4.40 | **2.70x** | 4.46 | 2.35x | 3.90 | 2.31x | 3.55 | **2.72x** | 4.24 |
| | Jakiro* | **3.04x** | **4.70** | 3.00x | 4.73 | **2.90x** | **5.01** | 2.61x | **4.77** | 2.04x | 3.94 | **2.44x** | **3.88** | 2.67x | **4.50** |
| L2 13B | Eagle1 | 2.59x | 3.69 | 3.03x | 4.23 | 2.69x | 3.81 | 2.54x | 3.64 | 2.34x | 3.45 | 2.37x | 3.34 | 2.59x | 3.69 |
| | Eagle2 | 2.82x | 4.61 | 3.37x | 5.40 | 2.97x | 4.79 | 2.75x | 4.52 | 2.44x | 4.20 | 2.54x | 4.08 | 2.81x | 4.60 |
| | Jakiro | 2.97x | 4.56 | **3.53x** | **5.29** | 3.19x | 4.79 | 2.95x | **4.52** | 2.55x | 4.22 | 2.76x | **4.15** | **2.99x** | **4.59** |
| | Jakiro* | **3.14x** | **4.72** | 2.75x | 4.03 | **3.33x** | **4.90** | 2.79x | 4.11 | **2.91x** | **4.50** | 2.77x | 4.10 | 2.95x | 4.39 |

**Results in A100-40G:**

*Table 6.* Speedup ratios and average acceptance lengths $\tau$ of different methods. V represents Vicuna, L2 represents LLaMA2-Chat. SpS denotes standard speculative sampling, with its draft model being Vicuna-68M. Methods like Medusa relax acceptance conditions under non-greedy settings, which do not guarantee lossless acceleration. Therefore, we do not compare Jakiro with these methods. Where the symbol * indicates that our method uses MoE-based weighted decoupling to construct the draft tree, as shown in Figure 3(2). Without the symbol *, it refers to the construction of the draft tree using weighted non-decoupling combined with the contrastive mechanism, as shown in Figure 5.

| Model | Method | MT-bench | | HumanEval | | GSM8K | | Alpaca | | CNN/DM | | Natural Ques. | | Mean | |
|---|---|---|---|---|---|---|---|---|---|---|---|---|---|---|---|
| | | Speedup | $\tau$ | Speedup | $\tau$ | Speedup | $\tau$ | Speedup | $\tau$ | Speedup | $\tau$ | Speedup | $\tau$ | Speedup | $\tau$ |
| | | | | | | | Temperature=0 | | | | | | | | |
| V 7B | Eagle1 | 2.84x | 3.98 | 3.23x | 4.30 | 3.18x | 3.99 | 2.75x | 3.87 | 2.48x | 3.42 | 2.42x | 3.21 | 2.82x | 3.80 |
| | Eagle2 | 3.29x | 4.98 | 3.71x | 5.35 | 3.15x | 4.94 | 2.97x | 4.86 | 2.62x | 4.11 | 2.46x | 3.82 | 3.03x | 4.68 |
| | Jakiro | **3.34x** | **5.03** | **3.81x** | **5.48** | **3.22x** | **4.98** | **3.06x** | **4.97** | **2.68x** | **4.13** | **2.50x** | **3.86** | **3.10x** | **4.74** |
| L2 7B | Eagle1 | 2.94x | 3.82 | 3.24x | 4.30 | 2.98x | 3.91 | 2.81x | 3.68 | 2.58x | 3.41 | 2.67x | 3.43 | 2.87x | 3.76 |
| | Eagle2 | **3.20x** | **4.70** | **3.63x** | **5.38** | **3.21x** | **4.77** | **3.12x** | **4.66** | **2.68x** | **4.10** | **2.77x** | **4.16** | **3.10x** | **4.63** |
| | Jakiro | 3.15x | 4.61 | 3.60x | 5.24 | 3.20x | 4.65 | 3.06x | 4.48 | 2.62x | 4.01 | 2.70x | 4.07 | 3.05x | 4.51 |
| V 13B | Eagle1 | 2.94x | 4.00 | 3.34x | 4.39 | 2.93x | 3.98 | 2.96x | 3.95 | 2.55x | 3.52 | 2.35x | 3.10 | 2.85x | 3.82 |
| | Eagle2 | 2.96x | **4.83** | 3.47x | 5.42 | 3.07x | 4.79 | 2.94x | **4.90** | 2.45x | **4.21** | 2.34x | **3.71** | 2.87x | **4.64** |
| | Jakiro | **3.05x** | 4.74 | **3.55x** | 5.32 | **3.23x** | **4.81** | **2.95x** | 4.70 | **2.56x** | 4.17 | **2.43x** | 3.67 | **2.96x** | 4.57 |
| L2 13B | Eagle1 | 2.97x | 3.93 | 3.39x | 4.52 | 3.20x | 4.02 | 3.00x | 3.83 | 2.65x | 3.58 | 2.61x | 3.46 | 2.97x | 3.89 |
| | Eagle2 | 3.00x | **4.75** | **3.57x** | **5.52** | 3.14x | **4.89** | 2.94x | **4.60** | 2.54x | **4.26** | 2.67x | **4.12** | 2.98x | **4.69** |
| | Jakiro | **3.03x** | 4.67 | 3.56x | 5.42 | **3.16x** | 4.79 | **2.96x** | 4.51 | **2.55x** | 4.16 | **2.67x** | 4.04 | **2.99x** | 4.60 |
| | | | | | | | Temperature=1 | | | | | | | | |
| V 7B | Eagle1 | 2.40x | 3.55 | 2.75x | 3.85 | 2.35x | 3.67 | 2.27x | 3.62 | 2.15x | 3.10 | 2.04x | 2.93 | 2.33x | 3.45 |
| | Eagle2 | 2.69x | 4.27 | 3.11x | 4.66 | 2.68x | 4.45 | 2.60x | 4.37 | 2.32x | 3.84 | 2.12x | 3.49 | 2.59x | 4.18 |
| | Jakiro | 2.66x | 4.31 | 2.96x | 4.57 | 2.69x | 4.49 | 2.50x | 4.30 | 2.20x | 3.85 | 2.09x | 3.52 | 2.52x | 4.17 |
| | Jakiro* | **3.86x** | **5.68** | **2.97x** | **4.88** | **3.32x** | **5.89** | **3.35x** | **5.54** | **3.29x** | **5.51** | **3.46x** | **5.14** | **3.38x** | **5.44** |
| L2 7B | Eagle1 | 2.52x | 3.64 | 2.79x | 4.05 | 2.56x | 3.77 | 2.47x | 3.58 | 2.15x | 3.15 | 2.30x | 3.23 | 2.46x | 3.57 |
| | Eagle2 | 2.95x | 4.49 | 3.39x | 5.04 | 3.05x | 4.69 | 2.94x | 4.56 | 2.42x | 3.94 | 2.48x | 4.06 | 2.87x | 4.46 |
| | Jakiro | 2.91x | 4.43 | **3.17x** | **4.87** | 2.90x | 4.57 | 2.76x | 4.38 | 2.32x | 3.89 | 2.37x | 3.98 | 2.74x | 4.35 |
| | Jakiro* | **3.36x** | **5.17** | 3.03x | 4.68 | **3.02x** | **4.65** | **3.31x** | **5.23** | **3.38x** | **5.36** | **3.30x** | **4.86** | **3.23x** | **4.99** |
| V 13B | Eagle1 | 2.57x | 3.60 | 2.71x | 3.92 | 2.64x | 3.78 | 2.35x | 3.73 | 2.14x | 3.30 | 2.01x | 3.01 | 2.40x | 3.56 |
| | Eagle2 | 2.62x | 4.31 | 3.02x | 4.80 | 2.75x | 4.46 | 2.61x | 4.54 | 2.28x | 4.00 | 2.17x | 3.49 | 2.57x | 4.27 |
| | Jakiro* | **3.00x** | **4.62** | **2.92x** | **4.76** | **2.93x** | **4.96** | **2.51x** | **4.84** | **1.96x** | **4.02** | **2.53x** | **4.03** | **2.64x** | **4.54** |
| L2 13B | Eagle1 | 2.73x | 3.67 | 3.08x | 4.27 | 2.68x | 3.80 | 2.52x | 3.66 | 2.22x | 3.37 | 2.42x | 3.36 | 2.61x | 3.69 |
| | Eagle2 | 2.75x | 4.58 | **3.31x** | 5.33 | 2.93x | 4.68 | 2.78x | **4.56** | 2.41x | 4.16 | 2.52x | 4.04 | 2.78x | **4.56** |
| | Jakiro* | **3.15x** | **4.67** | 2.93x | 4.10 | **3.43x** | **4.94** | **2.83x** | 4.14 | **2.88x** | **4.42** | **2.83x** | **4.16** | **3.01x** | 4.41 |

