# OpenReview forum: "Jakiro: Boosting Speculative Decoding with Decoupled Multi-Head via MoE"
_ICML.cc/2025/Conference — Submitted to ICML 2025_

### Official Review · Reviewer_ngDj · 2025-02-20

**Overall Recommendation:** 2

**Summary:**

This paper proposes Jakiro, a method that boosts the performance of speculative decoding for Large Language Model (LLM) inference acceleration.  Speculative decoding employs a smaller, faster "draft" model to predict upcoming tokens, which a larger "target" model then verifies.  Jakiro introduces two primary innovations: a dynamic decoupling mechanism using a Mixture of Experts (MoE) approach to enhance prediction diversity and a hybrid inference strategy combining autoregressive and parallel decoding.  This allows Jakiro to achieve state-of-the-art performance in speculative decoding.

**Claims And Evidence:**

The paper presents compelling ideas and generally provides evidence for its claims, there are certain areas where the support could be more robust and transparent:

Strengths:
Clear comparisons: The paper provides a detailed comparison with existing speculative decoding methods, including Medusa and Eagle, clearly outlining the limitations of these approaches and how Jakiro addresses them.
Experimental setup: The experimental setup is comprehensive, covering various models (Vicuna, LLaMA2-chat, LLaMA3-Instruct) and benchmark datasets (MT-bench, HumanEval, GSM8K, etc.).
Ablation studies: Ablation studies are conducted to analyze the impact of different components of Jakiro, such as the MoE settings and the contrastive mechanism, providing insights into their individual contributions.

Area for improvement:

Claim: "This suggests that Jakiro benefits from a more efficient drafting process that allows for longer and more stable sequences of tokens to be accepted, reducing the need for frequent re-sampling and minimizing the risk of errors during the inference process."
Issue: This claim says attributes speedup to "minimizing the risk of errors during the inference process". What errors are the authors talking about?

Issue: the work does not talk about the inference framework they used or if they did or did not use chunked prefill (a standard in modern production runtimes).

Overall, the paper presents a promising approach to speculative decoding with supporting evidence.

**Essential References Not Discussed:**

Non that I noticed

**Experimental Designs Or Analyses:**

Yes. As noted in the strengths in the "Claims And Evidence" section.

**Methods And Evaluation Criteria:**

Yes. As noted in the strengths above.

**Other Comments Or Suggestions:**

None

**Other Strengths And Weaknesses:**

Strength:
They are using the latest in architecture design (MoE architecture)

Weakness:
The result seems very incremental. They used an MoE (which most people should be doing now anyway) and got slightly better results at the cost of needing to store more draft model parameters.

**Questions For Authors:**

None

**Relation To Broader Scientific Literature:**

This work pushes the boundaries of Speculative Decoding by building on the latest in the space and integrating the MoE structure into it.

**Theoretical Claims:**

This paper does not present any formal theoretical proofs that would require checking for correctness. The claims made in the paper are primarily supported through empirical evidence obtained from experiments and ablation studies.

---

> ### Author Rebuttal · Authors · 2025-03-31
>
> > C1: This claim attributes speedup to "minimizing the risk of errors during the inference process". What errors are the authors talking about?
>
> Thank you for pointing out the ambiguity in our phrasing. The "risk of errors" refers to the probability that a token generated by the draft model is rejected by the target model during the validation phase of Speculative Decoding (SD). This rejection risk directly corresponds to the inverse of the acceptance rate of the draft model. When a draft token is rejected at a particular step:
>
> - All subsequent tokens in the speculative sequence are discarded.
> - The system must re-sample a new token from the target model, which resets the speculative decoding process for subsequent steps.
>
> A low acceptance rate (i.e., high rejection risk) leads to frequent re-sampling events, which disrupts the efficiency of SD. This is because:
>
> - **Computational Overhead:** Re-sampling from the target model incurs additional latency, compromising the speed gains from speculative decoding.
> - **Loss of Speculative Progress:** Discarded speculative tokens waste the computational effort invested in generating them.
>
> Jakiro’s design mitigates this risk by optimizing the draft model’s alignment with the target model, thereby increasing the acceptance rate. This results in fewer re-sampling events and more stable, longer sequences of accepted tokens, ultimately enhancing the overall speedup of the SD process.
>
> > Q1: The work does not talk about the inference framework they used or if they did or did not use chunked prefill. Overall, the paper presents a promising approach to speculative decoding with supporting evidence.
>
> Thanks for your valuable comment. Similar to mainstream SD methods (e.g., Medusa, EAGLE, Hydra), our Jakiro implementation relies solely on the PyTorch framework without additional acceleration architectures.
>
> Regarding "chunked prefill" (Agrawal et al., 2023), we didn't employ this technique because:
> - Our experiments used batch_size=1 by default, may refer to our response to Reviewer aZSm's comment W2.
> - The models could be fully loaded onto GPUs without memory constraints.
>
> We acknowledge that integrating "chunked prefill" could benefit larger models or batch_size>1 scenarios. We plan to explore such optimizations (e.g., vLLM/SGLang integration) in future work to further enhance Jakiro's performance.
>
> > W1: The result seems very incremental. They used an MoE (which most people should be doing now anyway) and got slightly better results at the cost of needing to store more draft model parameters.
>
> We appreciate this critical perspective but wish to clarify that characterizing our results as merely incremental may not be entirely accurate. While we do employ MoE, simply increasing model parameters does not guarantee improved speedup—expanding the draft model might improve acceptance rates but also introduces additional overhead, potentially reducing the speedup gain (as demonstrated in our ablation studies in Table 3).
>
> Furthermore, we optimize the MoE architecture by using slimmer MLP dimensions (detailed in the lower part of Figure 3 in this paper). To verify that Jakiro introduces minimal additional parameters and almost no extra memory overhead during inference, please refer to our response to Reviewer LwQj’s comment W2.
>
> We maintain that integrating MoE into speculative decoding remains an innovative and non-trivial contribution.

---

> > ### Comment · Reviewer_ngDj · 2025-04-08
> >
> > i have read the response and will keep my rating

---

### Official Review · Reviewer_Ms2i · 2025-03-12

**Overall Recommendation:** 4

**Summary:**

This paper presents Jakiro, which utilizes the MoE technique to do two-token-ahead parallel decoding to enhance the diversity of draft model prediction. Upon the framework of EAGLE, Jakiro replaces the MLP layer of EAGLE drafter with an MoE layer consisting of a router and several experts.

The authors also integrated popular contrastive decoding techniques at the feature level (i.e., hidden states before LLM head), where authors claimed to achieve further improvement on greedy decoding scenarios.

**Claims And Evidence:**

- Diversity of draft tokens increased: per my understanding, authors have a strong claim that the Jakiro-style method significantly increases the “diversity” of draft tokens. However, other than the drafted tokens are now from 2 sets of features from selected experts, I did not see any signs of diversity in the drafted tokens. I would rather ask the authors to give a clear definition of the diversity of draft tokens and then quantize it with such a definition.
    - If the authors only refer to the high performance in high-temperature decoding, I wouldn’t choose the term diversity if I were in their shoes.

- The motivation for contrastively decoding the token with top-2 selected experts is strange. The authors have not explained the reason, and it is not obvious to tell. I would consider using the features from the most activated expert and least activated experts to do contrastive decoding, as they could serve the role of strong and weak models in the original contrastive decoding setups.

**Essential References Not Discussed:**

The parallel decoding usage of SPD is a crucial contribution of Jakiro, however, the authors missed some of the recent works in parallel SPD including:

BiTA: Bi-Directional Tuning for Lossless Acceleration in Large Language Models,
Parallel Decoding via Hidden Transfer for Lossless Large Language Model Acceleration,
ParallelSpec: Parallel Drafter for Efficient Speculative Decoding

**Experimental Designs Or Analyses:**

The experiments are sound and solid.

**Methods And Evaluation Criteria:**

The datasets and metrics used in this paper are quite established and standard.

**Other Comments Or Suggestions:**

Typos:
L311: 77B -> 70B

**Other Strengths And Weaknesses:**

Strengths:
- Jakiro achieved SOTA results with both greedy and non-greedy decoding modes, which is impressive as decoding with temperature has always been challenging for speculative decoding methods.
- Appendix A.2 illustrates the speedup ratio on different devices, confirming the universal applicability of Jakiro.

Weaknesses:
- Table 3 seems to tell regretting facts that although introducing the MoE mechanism helps increase the average acceptance length, the speed-up ratio suffers as the number of experts increases. Per my understanding, N=K=2 means a simple EAGLE-style implementation with learnable ratios and contrastive decoding between two heads.

**Questions For Authors:**

See Claims And Evidence.

**Relation To Broader Scientific Literature:**

N/A

**Theoretical Claims:**

No proofs for theoretical claims are provided in the paper.

---

> ### Author Rebuttal · Authors · 2025-03-31
>
> > C1: The authors should give a clear definition of the diversity of draft tokens and then quantify it with such a definition.
>
> Thanks for your thoughtful feedback. Building upon our previous response to Reviewer LwQj (W1), as illustrated in the figure, our Jakiro model, even with just two MOE heads, can generate richer tokens.
>
> To rigorously address your query on defining and quantifying diversity, we propose a composite metric that evaluates three dimensions:
>
> - **Generation Richness (G):** The number of unique draft tokens, used to evaluate the diversity of the drafting phase.
>
> - **Selection Effectiveness (S):** The number of unique accepted tokens, used to evaluate the diversity of the verification phase.
>
> - **Exploration Depth (E):** The total number of tokens generated in the final output, prioritizing deeper exploration.
>
> The diversity metric is formalized as:
> $$
> \text{Diversity} = G + S +E = \frac {\sum ^{i}_{N} {(\alpha \cdot \ln{g_i}  + \beta \cdot \ln{s_i} + \gamma \cdot \ln{e_i} )}} {N}
> $$
> where $g_i$, $s_i$, and $e_i$ represent the Generation Richness, Selection Effectiveness, and Exploration Depth for each round of dialogue, respectively, and $N$ denotes the total number of dialogue rounds. The empirically validated weights are: $\alpha=0.4,\ \beta=0.4,\ \gamma=0.2$.
>
> Benchmark results on mt_bench (Vicuna-7B, T=1, top-k=10):
>  Method        | Avg. G ↑ | Avg. S ↑ | Avg. E ↑ | Diversity ↑
> :-------------:|:--------:|:--------:|:--------:|:-----------:
>  Eagle2 (Dense)| 1234     | 206      | 789      | 6.3
>  Jakiro (MoE)  | 4830     | 246      | 865      | **7.0**
>
> Jakiro achieves 11% higher diversity than the Eagle, driven by its ability to generate more unique tokens, maintain high acceptance rates, and explore longer sequences.
>
> > C2: The motivation for contrastively decoding with top-2 selected experts. Consider using the features from the most activated expert and least activated experts to do contrastive decoding.
>
> Thank you for raising this important point. The rationale for using top-2 activated experts in contrastive decoding:
>
> - **Baseline Configuration:** Our initial implementation followed the mainstream MoE framework, where the top-2 experts (ranked by routing weights) are selected for token generation.
>
> - **Empirical Optimization:** As shown in Table 3 of the paper, using 2 experts achieved the optimal speedup.
>
> To rigorously evaluate our design, we conduct additional Vicuna 7B experiments comparing two strategies on A40:
>
> |   | Strategy   | #Expert | MT-bench Speedup | MT-bench τ | HumanEval Speedup | HumanEval τ |
> |:---:|:----------:|:-------:|:----------------:|:----------:|:-----------------:|:-----------:|
> |   | top-2      | 5       | 2.59x            | 5.13       | 2.95x             | 5.60        |
> |   | top-bottom | 5       | 2.45x (-5.4%)    | 4.82       | 2.80x (-5.1%)     | 5.20        |
> |   | top-2      | 4       | 2.65x            | 5.09       | 3.02x             | 5.54        |
> |   | top-bottom | 4       | 2.50x (-5.7%)    | 4.75       | 2.85x (-5.6%)     | 5.12        |
>
> We fully agree with the reviewer on the theoretical significance of exploring "strong-weak expert contrastive decoding." However, our experiments reveal that low-confidence experts (e.g., bottom-1) may introduce noise, leading to degraded output quality. We analyze that this discrepancy from the original contrastive decoding conclusions (which compare logits from strong/weak models) might stem from: our contrastive decoding mechanism compares **hidden states** of experts. This architectural difference could explain why the "strong-weak expert" paradigm behaves differently in our framework.
>
> > R1: Missed some of the recent works in parallel SPD.
>
> We will discuss recent parallel SPD works like BiTA, Parallel Decoding via Hidden Transfer, and ParallelSpec in the revised manuscript.
>
> > W1:  The speed-up ratio suffers as the number of experts increases in Table 3. N=K=2 means a simple EAGLE-style implementation with two heads.
>
> We would like to clarify two key points:
> 1. It fundamentally differs from EAGLE in that our Jakiro uses dynamic **router-based** expert selection for *autoregressive phases* (Stages 1-4) and employs contrastive decoding only in the final *parallel phase* (Stages 5-6).
> 2. In smaller models (e.g., Vicuna-7B), increasing N introduces **additional computational costs** (e.g., router computation), which slows down the autoregressive phase.
>
> To address these trade-offs, we propose the following directions:
>
> - **Efficiency Optimization for Small Models:** Explore lightweight routing mechanisms or parameter-sharing techniques to reduce N’s overhead.
> - **Dynamic N/K Adjustment:** Adaptively set N and K based on task complexity (e.g., higher N for complex tasks, lower N for simplicity).
>
> Our MoE design inherently supports scalable improvements for large models (e.g., DeepSeek-V3-671B), and the draft model’s overhead becomes negligible relative to total computation (Experts N>2).

---

> > ### Comment · Reviewer_Ms2i · 2025-04-02
> >
> > Thanks for the detailed rebuttal. I keep my `accept` rating. I believe Jakiro is a good piece of work and good luck!

---

### Official Review · Reviewer_aZSm · 2025-03-14

**Overall Recommendation:** 2

**Summary:**

This paper proposes Jakiro, which leverages Mixture of Experts (MoE), where independent experts generate diverse predictions, effectively decoupling correlations among candidates. It demonstrates universal improvements across multiple different benchmarks.

**Claims And Evidence:**

Yes, LLM acceleration is a very important topic for current applications. However, the authors should test over more advanced settings like flash decoding and bsz > 1.

**Essential References Not Discussed:**

All the essential references are clearly discussed.

**Experimental Designs Or Analyses:**

The authors should test over more advanced settings like flash decoding.

The authors should test over throughput with different bsz.

The authors should also add models like Qwen which are better at math and coding.

**Methods And Evaluation Criteria:**

The authors should also report thoughput result.

**Other Comments Or Suggestions:**

see cons.

**Other Strengths And Weaknesses:**

Advantages:

1. The paper is well-written.

2. I think the idea is neat. Intuitively, both MoE and semi-autoregressive should work.

3. The experimental results strongly support the effectiveness of this method.

Disadvantages:

1. The paper lacks a direct comparison with a strong baseline using flashdecoding.

2. According to the ablation study, the contrastive loss seems not to work. If the improvement is marginal, in my opinion, the authors can omit this part to keep the method simple.

3. The authors should test over more advanced settings like flash decoding.

4.  The authors should test over throughput with different bsz.

If the authors add more experiments over 3 & 4, I will increase score from 2 to 3.

**Questions For Authors:**

see cons.

**Relation To Broader Scientific Literature:**

NA. The idea of this paper is specifically designed for speculative decoding.

**Theoretical Claims:**

I did not see any proof.

---

> ### Author Rebuttal · Authors · 2025-03-31
>
> > W1: The authors should test over more advanced settings like flash decoding.
>
> Thank you for the feedback. We first clarify that Flash Decoding (FD) and speculative decoding (SD) operate at different optimization levels but can be effectively combined for greater efficiency.
>
> **Flash Decoding (Dao et al., 2023):**
>
> A system-level optimization that accelerates attention computation for long-context processing through:
> - Memory-efficient tiling for KV caching
> - Parallel reduction across sequence chunks
>
> **Speculative Decoding (Leviathan et al., 2023):**
>
> An algorithmic innovation leveraging the computation-memory gap in LLM inference:
> - Draft model proposes candidate tokens (reducing generation steps)
> - Target model verifies in parallel (exploiting GPU underutilization)
>
> By conducting Vicuna 7B experiments on A100 under T=0 (Refer to FlashDecoding++, Hong et al., 2023):
>
>  Method        | mt_bench | humaneval | gsm8k | Avg
> ---------------|----------|-----------|-------|-------
>  Baseline + FD | 2.05x    | 2.06x     | 2.10x | 2.07x
>  Jakiro        | 3.34x    | 3.81x     | 3.22x | **3.46x**
>
> > W2: The authors should test over throughput with different bsz.
>
> Similarly, we conducted experiments on "Batch Sizes > 1" (T=0) with Vicuna 7B on A40, and the results are shown in the table below. These results validate that the speedup of our method increases as the batch size grows, but the gains diminish gradually.
>
>  BS | mt_bench | humaneval | gsm8k | Avg
> ----|----------|-----------|-------|-------
>  1  | 3.02x    | 3.40x     | 3.08x | 3.17x
>  2  | 2.98x    | 3.35x     | 3.05x | 3.13x
>  4  | 2.85x    | 3.25x     | 2.95x | 3.02x
>  8  | 2.72x    | 3.10x     | 2.82x | 2.88x
>
> Additional Clarification on Applying SD to **Batch Sizes > 1**:
>
> - **Performance Degradation with Larger Batches**: Simply increasing the batch size during SD inference shifts the problem nature from memory-bound to compute-bound, leading to diminishing returns or even negative impacts as batch sizes grow (MagicDec, Sadhukhan et al., 2025).
>
> - **Practical Focus on Batch Size = 1**: Current SD optimizations prioritize batch size = 1 due to
>
>    - *Sequence Length Variability*: Divergent acceptance rates across sequences in a batch result in varying candidate sequence lengths after the drafting stage.
>
>    - *Verification Overhead*: This inconsistency increases computational costs (or latency) during the parallel verification phase of the target model.
>
> **Research Status & Outlook:**
> Existing SD implementations (e.g., the seminal SD work, Medusa, EAGLE, Hydra) predominantly target batch size = 1. Though optimizing for batch sizes > 1 remains an open challenge, it represents a promising direction for future research.
>
> > W3: If the improvement of 'contrastive loss' is marginal, the authors can omit this part to keep the method simple.
>
> We appreciate the reviewer's observation regarding the contrastive loss. While the absolute improvement may appear modest, the contrastive loss is architecturally indispensable for maintaining the system's end-to-end performance. Specifically on GSM8K of Table 4 in this paper, the contrastive loss contributes to a measurable Speedup enhancement (3.05x → 3.11x). Its critical role lies in:
>
> - Optimizing later-stage parallel decoding efficiency
> - Preserving the acceptance rate during initial auto-regressive phases
>
> Of course, this part can be omitted to maintain method simplicity if a certain performance loss is acceptable.
>
> > S1: The authors should also add models like Qwen with better at math and coding.
>
> Thanks for your valuable suggestion. We have added comprehensive experiments on Qwen2-7B-Instruct in A40 (Speedup: λ, Average accepted length: τ):
>
>  Method | T | MT-bench λ | MT-bench τ | HumanEval λ | HumanEval τ | GSM8K λ | GSM8K τ | Alpaca λ | Alpaca τ | CNN/DM λ | CNN/DM τ | Natural Ques. λ | Natural Ques. τ | Mean λ    | Mean τ
> --------|---|------------|------------|-------------|-------------|---------|---------|----------|----------|----------|----------|-----------------|-----------------|-----------|--------
>  Eagle2 | 0 | 2.13       | 4.16       | 2.23        | 4.18        | 2.05    | 3.93    | 1.70     | 3.30     | 1.75     | 3.43     | 1.44            | 2.73            | 1.88x     | 3.62
>  Jakiro | 0 | 2.28       | 4.20       | 2.36        | 4.20        | 2.16    | 3.98    | 1.82     | 3.35     | 1.88     | 3.46     | 1.53            | 2.75            | **2.01x** | 3.64
>  Eagle2 | 1 | 1.61       | 3.18       | 1.69        | 3.28        | 1.75    | 3.41    | 1.30     | 2.56     | 1.18     | 2.36     | 1.13            | 2.19            | 1.44x     | 2.83
>  Jakiro | 1 | 1.72       | 3.20       | 1.81        | 3.30        | 1.85    | 3.45    | 1.38     | 2.60     | 1.25     | 2.40     | 1.20            | 2.25            | **1.54x** | 2.85
>
>
> These additions highlight Jakiro’s adaptability to specialized LLMs.

---

> > ### Comment · Reviewer_aZSm · 2025-04-02
> >
> > Thank you for the detailed response and the extensive experiments. I appreciate the effort you’ve put in. I would like to raise a few additional points and clarify my expectations for the final round:
> >
> > ***1. Flash Decoding Compatibility with Tree Attention:***
> > In W1, I asked about integrating Flash Decoding (FD) into your method. I would like to re-emphasize that FD is now a default system-level optimization with strong and stable acceleration benefits, and any method aiming for practical deployment must demonstrate compatibility with it.
> > Your Tree Attention appears promising, but if it cannot be combined with FD, that significantly limits its real-world applicability. In fact, I have raised this exact concern in all speculative decoding papers I’ve reviewed.
> > I do not consider the argument that FD and speculative decoding are "orthogonal" to be sufficient—FD is the default, and speculative decoding must work on top of it to be practically useful.
> > If Tree Attention cannot be made compatible with FD, **I will not increase the score**. I encourage you to include substantial experiments showing this compatibility, especially with the **group query attention models**.  Additionally, speculative decoding without tree attention is compatible with FD, and I have implemented one, which is faster but needs fewer flops than the tree attention one. **You should also include this setting as a baseline.**
> >
> > ***2. Throughput Under Batch Sizes > 1:***
> > Thank you for providing batch size > 1 results. However, the maximum batch size is too small to demonstrate practical throughput performance, and A40 is not ideal for this purpose. I strongly recommend testing on A100 (80GB), where throughput is commonly measured in thousands of tokens per second.
> > In addition, I’m now requesting you to include GQA models like Qwen in your FD + batch size > 1 experiments to better validate Jakiro’s generality under modern architectures. (Note: this requirement is new and was not part of my initial comments.)
> >
> > ***3. Contrastive Loss Impact:***
> > The results confirm my intuition.
> >
> > ***Final Note:***
> > Please note that this is the **final** opportunity to respond, so I recommend reporting as many models' results as possible. You can focus on the single dataset SpecBench to simplify evaluation. That said, I will be monitoring your updates closely, as soon as you upload new results, I will review them promptly and revise my score if the key issues (particularly FD compatibility and max throughput) are convincingly addressed.
> >
> > Thank you again for your work, and I look forward to your final response.

---

> > > ### Author Response · Authors · 2025-04-08
> > >
> > > Thanks for your responsible review.
> > >
> > > (1) To be honest, the validation of applying Flash Decoding (FD) to speculative decoding (SD) with Tree Attention structure is indeed a challenging issue. While you emphasized that experimental validation of FD combined with SD is necessary to prove Jakiro's general applicability, to our knowledge, there appears to be no open-source Tree Attention-based speculative decoding method that has implemented this integration. If any exists, we sincerely hope you could inform us for future study. We attempted to integrate vLLM's technology (which claims to use FD) during the rebuttal but found they haven't applied it to Tree Attention. We think this is more of an engineering optimization issue than this paper's core focus. Although our paper initially received modest scores with slim acceptance chances, we also attempt to address your concerns responsibly.
> > >
> > > (2) Regarding your recommendation to conduct experiments with larger batch sizes on A100-80GB: Our lab and surrounding facilities lack this GPU model. We rented two instances on the AutoDL platform but observed unstable test results. The following data shows average results from 5 runs on Qwen2-7B-Instruct with GQA (for reference only):
> > >
> > >  BS | mt_bench(T=0)  | mt_bench(T=1)
> > > ----|----------------|---------------
> > >  1  | 2.75x          | 2.11x
> > >  2  | 2.93x          | 2.28x
> > >  4  | 2.78x          | 2.08x
> > >  8  | 2.71x          | 2.03x
> > >  16 | 2.64x          | 1.96x
> > >  32 | 2.48x          | 1.89x
> > >
> > > Finally, we sincerely appreciate your feedback, though the requirements are indeed challenging. Even if this paper is not accepted, we will continue to explore the efficient integration of Jakiro with FlashDecoding technology to make it a more practical speculative decoding solution.

---

### Official Review · Reviewer_LwQj · 2025-03-14

**Overall Recommendation:** 2

**Summary:**

The paper claims that Jakiro improves speculative decoding by leveraging the Mixture of Experts (MoE) for dynamic decoupling and introduces a hybrid inference strategy that combines autoregressive decoding with parallel decoding in the last steps. The authors also claim that Jakiro achieves state-of-the-art performance in speculative decoding by significantly improving prediction accuracy and inference speed.

**Claims And Evidence:**

The claims are validated by experimental results.

**Essential References Not Discussed:**

N/A

**Experimental Designs Or Analyses:**

The experimental design is solid while lacking specific parts that can further improve the integrity. See weaknesses.

**Methods And Evaluation Criteria:**

The proposed method can tackle the challenges identified by the authors.

**Other Comments Or Suggestions:**

The paper lacks an overview figure for the proposed framework.

**Other Strengths And Weaknesses:**

Strengths:

Novelty: This work effectively decouples token dependencies within the draft tree, improving token diversity and verification accuracy.
Experiments: The authors considered multiple model scales (7B to 70B) and diverse task benchmarks to provide strong empirical validation.

Weaknesses:

The authors claim that their work has improved diversity. However, the corresponding analysis is missing in the experiments.

While MoE increases speedup, there is no analysis of efficiency or memory overhead compared to non-MoE speculative methods.

The proposed method seems to be incorporating the architecture of MoE into speculative decoding. This may seem to lack novelty, and the authors should provide more justification for this design regarding its novelty and meaningfulness.

**Questions For Authors:**

1. In Figure 1. Comparison of different speculative decoding methods, the authors state that this includes multiple speculative decoding methods. However, in the (a) part of the figure, it says it is speculative decoding. How is this particular method related to any other three methods in the figure, if all of them are considered speculative decoding in the caption, I suppose?

2. What does the name mean? The method name Jakiro seems not to be any particular machine learning algorithms or frameworks.

**Relation To Broader Scientific Literature:**

This work meaningfully advances speculative decoding research

**Theoretical Claims:**

No theoretical claims or proofs were provided.

---

> ### Author Rebuttal · Authors · 2025-03-31
>
> > W1: The analysis of diversity is missing in the experiments.
>
> To validate that our Jakiro method enhances the diversity of speculative sampling, we conduct a comparative analysis against Eagle2 (with temperature=1). Using a prompt from mt_bench: "Compose an engaging travel blog post about a recent trip to Hawaii, highlighting cultural experiences and must-see attractions." We statistically evaluate the responses averaged over 3 trials. The figure (https://anonymous.4open.science/r/Jakiro_rebuttal-C4C2/D.png) visualizes the frequency heatmap of the top-10 drafted tokens (default setting) sampled during the entire response generation. Tokens with frequencies below 10 were filtered out for clearer visualization.
>
> The results show:
>
> - **Higher Token Quantity**: Under identical experimental conditions, Jakiro generates significantly more draft tokens (vertical axis of the figure).
> - **Broader Semantic Coverage**: The generated tokens occupy a wider span in the semantic embedding space (horizontal axis), reflecting richer topical diversity.
>
> Additionally, we also provide the statistical results of accepted tokens for Jakiro* and Eagle2 on the MT-bench dataset. See https://anonymous.4open.science/r/Jakiro_rebuttal-C4C2/C2.png, which further highlights the diversity of our Jakiro.
>
> > W2: No efficiency or memory overhead analysis compared to non-MoE speculative methods.
>
> We sincerely appreciate this insightful question. Here is our Memory Overhead Analysis:
>
> *Table: Hardware Metrics on A40-45GB of Vicuna 7B* (measured by `nvidia-smi` as shown in https://anonymous.4open.science/r/Jakiro_rebuttal-C4C2/M.png)
>  Metric             | Dense | MoE-2 | Δ
> :------------------:|:-----:|:-----:|:------:
>  Mem (GB)           | 15.08 | 15.26 | +0.6%
>  Latency (ms/token) | 13.13 | 11.74 | -10.6%
>
> Since Jakiro employs a lighter-weight MLP than Eagle, with merely 0.6% additional memory usage, the MoE-2 variant delivers a 10.6% speedup in token generation latency, showing highly efficient computation-memory scaling.
>
> > W3: The justification of Jakiro's design regarding its novelty and meaningfulness.
>
> Jakiro's novelty lies not in simply applying MoE to speculative decoding but in developing a *dynamic decoupling framework* that fundamentally addresses a newly identified bottleneck. Here are key justifications:
>
> - **Problem Innovation**
>    Prior works focus on temporal decoupling (Eagle) or multi-head prediction (Medusa) but overlook *in-step candidate correlation*. This intrinsic limitation motivates our MoE-based decoupling at the *intra-step level*.
>
> - **Architectural Novelty**
>    Compared to standard MoE applications:
>    - *Dynamic Routing*: Experts specialize in two-branch token speculative decoding.
>    - *Semi-autoregressive:* Combines autoregressive decoding for early tokens and parallel decoding for later stages.
>    - *Contrastive MoE*: The first to apply a contrastive mechanism between activated experts.
>
> > S1: The paper lacks an overview figure for the proposed framework.
>
> Thanks for your suggestion. We will include the framework diagram (https://anonymous.4open.science/r/Jakiro_rebuttal-C4C2/F.png) in the revised version.
>
> > Q1: The relationship between the particular method in (a) part of Figure 1 and the other three methods.
>
> Figure 1 (a) presents the baseline method of classical speculative decoding (i.e., SpS in Table 1 of this paper), which serves as the comparative reference for the improved approaches in (b) Medusa, (c) Eagle, and (d) Jakiro. We will explicitly annotate this relationship in the revised version.
>
> > Q2: What does Jakiro's name mean?
>
> The name 'Jakiro' is inspired by the twin-headed dragon character from the DOTA game, symbolizing our method's dual-expert architecture where two specialized activated heads collaboratively generate diverse token predictions.

---

### Decision · Program_Chairs · 2025-05-01

**Decision:**

Reject

**Comment:**

Summary:

This paper proposes Jakiro, enhancing speculative decoding for large language models by using a Mixture of Experts (MoE) for dynamic decoupling and a hybrid inference strategy combining autoregressive and parallel decoding. Jakiro achieves state-of-the-art performance, improving prediction accuracy and inference speed across various benchmarks.


Strengths:

1. Jakiro achieves state-of-the-art performance on difference devices.

Weaknesses:

1. The paper lacks justification and convincing (theoretical and experimental) analysis why MoE would lead to more diversity benefits.

2. Ablation study compared with non-MoE method in terms of efficiency or memory overhead is missing.

3. The largest batch size in the supplementary experiments of rebuttal is modest and might be too small to show the advantage of the proposed method in the large batch settings (the authors extend it to 32 in the second round of rebuttal).

4. Two reviewers have concerns on the the motivation and effectiveness of contrastive decoding.

5. Some references are missing.

Other comments:
AC does not think it is a must to combine this paper with FastDecoding or chunked prefill.

After the rebuttal and discussion, unfortunately, 3 out of 4 reviewers vote for (weak) rejection while only one reviewer votes for acceptance. Given the majority vote, AC would recommend rejection.